# The Phytochemistry and Pharmacology of Onocleaceae Plants: *Pentarhizidium orientale*, *Pentarhizidium intermedium*, and *Matteuccia struthiopteris*—A Review

**DOI:** 10.3390/plants14111608

**Published:** 2025-05-25

**Authors:** Jungmoo Huh

**Affiliations:** Research Institute of Pharmaceutical Sciences, College of Pharmacy, Seoul National University, Seoul 08826, Republic of Korea; goodhjm112@snu.ac.kr

**Keywords:** Onocleaceae, *Matteuccia struthiopteris*, *Pentarhizidium orientale*, *Pentarhizidium intermedium*, phytochemistry, pharmacology

## Abstract

The Onocleaceae family, a small group within the Pteridophytes, comprises four genera, but has been phytochemically studied mainly for *Pentarhizidium orientale*, *Pentarhizidium intermedium*, and *Matteuccia struthiopteris*. To date, a total of 91 compounds have been isolated from these three species, including 15 flavonoids, 48 flavonoid glycosides, 6 stilbenes, 4 isocoumarins, 2 phthalides, 3 chromones, 2 lignan glycosides, 8 isoprenoid derivatives, and 3 phenolic compounds. Notably, most flavonoids and flavonoid glycosides possess *C*-methyl groups at the C-6 and/or C-8 positions, with several conjugated to (*S*)-3-hydroxy-3-methylglutaryl (HMG) moieties. Although not all isolates have been evaluated for their pharmacological activities, several compounds have demonstrated bioactivities such as antiviral, anti-inflammatory, α-glucosidase inhibitory, aldose reductase inhibitory, and antioxidant effects.

## 1. Introduction

Among the pteridophytes, Onocleaceae represents a small but distinct family, classified into four genera: *Matteuccia*, *Onocleopsis*, *Onoclea*, and *Pentarhizidium* [1]. Of these, the genus *Pentarhizidium* comprises two species, *P*. *orientale* (Hook.) Hayata (syn. *M. orientalis*) and *P*. *intermedium* (C.Chr.) Hayata (syn. *M*. *intermedia*). These species were originally classified under the genus *Matteuccia* but were subsequently reassigned to a distinct genus. Interestingly, all these Onocleaceae plants are distributed across the Northern hemisphere: *P*. *orientale*, *O*. *sensibilis* var. *interrupta* are found in East Asia such as China, Korea, and Japan; *P*. *intermedium* is native to China; *O*. *hintonii* occurs in Mexico; and *M*. *struthiopteris* is widely distributed across the mid-latitudes of the Northern hemisphere [1,2].

According to the World Flora Online Plant List database (WFO Plant List, https://wfoplantlist.org) (accessed on 15 March 2025) [3], *Matteuccia* genus includes two species (*M*. *pensylvanica* and *M*. *struthiopteris*), *Onocleopsis* contains one species (*O*. *hintonii*), *Pentarhizidium* genus comprises two species (*P*. *orientale*, *P*. *intermedium*), and *Onoclea* genus encompasses three species (*O*. *pensylvatica*, *O*. *sensibilis*, *O*. *struthiopteris*).

Despite the taxonomic diversity of this family, to date, phytochemical studies have been conducted only on *M*. *struthiopteris*, *P*. *orientale*, and *P*. *intermedium*, with no such research reported for the other species within Onocleaceae. Therefore, this review aims to comprehensively summarize the current knowledge on phytochemistry, pharmacological activities of *P*. *orientale*, *P*. *intermedium*, and *M*. *struthiopteris*, thereby providing a foundation for future studies on the Onocleaceae family.

## 2. Results and Discussion

### 2.1. Botany

The scientific name of *P*. *orientale*, *P*. *intermedium*, and *M*. *struthiopteris* have been checked in the World Flora Online Plant List (https://wfoplantlist.org) (accessed on 15 March 2025). *P*. *orientale* is a perennial pteridophyte that grows in mountain forest (Figure 1A). The rhizome of *P*. *orientale* is thick and covered with scales. The plant height ranges from 70 to 150 cm and *P*. *orientale* has two types of fronds. The sterile fronds are larger than fertile fronds and its leaf blades are hairless, measuring 30 to 50 cm long with 20 to 30 cm width. Fertile fronds, which are dark brown in color, arise between the sterile fronds, and they grow 30 to 70 cm long, and have long petioles [4].

The rhizome of *P*. *intermedium* shows a dark brown color, is short and robust, and it grows erect. The end of the rhizome is covered with dark brown scales. Same as *P*. *orientale*, it has two types of fronds. The length of sterile frond ranges from 40 to 60 cm and the width appears from 15 to 25 cm, and that of fertile frond have 30 to 45 cm long and 8–15 cm wide [5].

*M*. *struthiopteris*, known as ostrich fern, is a perennial deciduous pteridophyte that grows in sunny wetlands within forests (Figure 1B). It reaches a height of 30–100 cm long. The rhizome of *M. struthiopteris* is short and grows upright. The scales are narrowly lanceolate, measuring 10–15 mm in length, and are membranous. The petiole of the sterile fronds ranges from 6 to 10 cm long and bears reddish-brown scales. The length of the leaf blade is 40–60 cm and the width of the leaf blade is 15–25 cm. The fertile fronds have lanceolate and 15–30 cm long leaf blades. The pinnae roll backward, and sporangia clusters are covered by protective membranes. The spores are produced between September and November [4].

### 2.2. Phytochemistry

In the present study, a total of 91 compounds were isolated from the root, aerial parts, or rhizome of *P*. *orientale*, *P. intermedium*, and *M. struthiopteris*. Among the isolated compounds, two-thirds of the compounds were flavonoids (**1**–**15**) and flavonoid glycoside derivatives (**16**–**63**). Furthermore, several classes of compounds were also reported such as six stilbene derivatives (**64**–**69**), four isocoumarins (**70**–**73**), two phthalides (**74** and **75**), three chromones (**76**–**78**), two lignan glycosides (**79** and **80**), eight isoprenoid derivatives (**81**–**88**), and three phenolics (**89**–**91**). The distribution of compound classes identified from *P. orientale*, *P. intermedium*, and *M. struthiopteris* is summarized in Figure 2.

#### 2.2.1. Flavonoids and Flavonoid Glycosides

Until now, 15 flavonoids (**1**–**15**) and 48 flavonoid glycosides (**16**–**63**) have been isolated and identified from *P*. *orientale*, *P. intermedium*, and *M. struthiopteris*. Notably, many flavonoids and flavonoid glycosides exhibited *C*-methyl at C-6 and/or C-8 position. Furthermore, 17 flavonoid glycosides (**47**–**63**) had (*S*)-3-hydroxy-3-methylglutaryl (HMG) moiety. Although the C-3 position of HMG is naturally biosynthesized as an *S* configuration [6], the authors confirmed this by performing chemical derivatization according to Hattori et al. [7] and determined the absolute configuration of HMG moiety.

Two *C*-methylated flavanone, demethoxymatteucinol (**1**) and matteucinol (**2**) were reported from *P*. *orientale* [8,9,10,11,12], *P. intermedium* [13], and *M. Struthiopteris* [14]. Matteucin (**3**), and farrerol (**4**) were isolated from the rhizome of *P*. *orientale* [8,12] and *P*. *intermedium* [13]. 2′-hydroxymatteucinol (**5**) is only identified from *P*. *orientale* [9,10,11], whereas 3′-hydroxymatteucinol (**8**) is solely confirmed in *P. intermedium* [13]. Methoxymatteucin (**6**) and cyrtominetin (**7**) were discovered from *P. intermedium* in 2019 [13], and methoxymatteucin (**6**) was also isolated and identified from *P*. *orientale* [8,12] and *M. struthiopteris* [14]. Huh et al. [12] reported that 3′-hydroxy-5′-methoxy 6,8-dimethyl huazhongilexone (**9**) and naringenin (**12**) were isolated from *P*. *orientale*. Ophiofolius A (**10**) was found in *M. struthiopteris*, together with matteuorien (**11**) and protoapigenone (**15**) [15]. Matteuorien (**11**) was also isolated from *P*. *orientale* reported by Basnet et al. [10]. From the *P*. *intermedium* and *M. struthiopteris*, 5,7-dihydroxy-4′-methoxy-6-methyl flavanone (**13**) was isolated and characterized [13,16] and demethylmatteucinol (**14**) was the only one confirmed in *P*. *intermedium* [13]. The chemical structures of all flavonoids are shown in Figure 3 and Table 1.

Demethoxymatteucinol 7-*O*-β-D-glucoside (**16**) and matteucinol 7-*O*-β-D-glucopyranoside (**17**) were first isolated from *P*. *orientale* and reported by Basnet et al. [10] in 1995, and Huh et al. [12], Li et al. [13], and Zhang et al. [16] also confirmed these two compounds in *P*. *orientale*, *P. intermedium*, and *M. struthiopteris*, respectively. Huh et al. [12] identified matteucin 7-*O*-β-D-glucopyranoside (**18**), myrciacitrin II (methoxymatteucin 7-*O*-β-D-glucopyranoside) (**20**), and matteuorien 7-*O*-β-D-glucopyranoside (**21**) from the rhizome of *P*. *orientale* in 2017, and Li et al. [13] obtained farrerol 7-*O*-β-D-glucopyranoside (**19**) and myrciacitrin II (**20**) from the 60% ethanolic extract of the rhizomes of *P*. *intermedium*.

Matteuflavosides A–J (**22**–**29**, **43**, **44**) were isolated and confirmed from the rhizomes of *M. struthiopteris* [15] and only matteuflavoside G (**26**) was also found in *P*. *intermedium* in 2019 [13]. Two acetylated flavonoid glycosides, (2*S*)-5,7-dihydroxy-6,8-dimethyl-dihydroflavone-7-*O*-(6″-*O*-acetyl)-β-D-glucopyranoside (**45**) and (2*S*)-5,7-dihydroxy-6,8-dimethyl-4′-methoxydihydroflavone-7-*O*-(6″-*O*-acetyl)-β-D-glucopyranoside (**46**), were found in *M*. *struthiopteris* in 2013 [16]. Eleven matteuorienates (**47**–**57**) and six matteuinterates (**58**–**63**) are composed of a flavonoid aglycone, a sugar moiety, and an HMG group. Kadota et al. [11] first discovered matteuorienates A and B (**47** and **48**) from *P*. *orientale* in 1994, and Basnet et al. [10] reported matteuorienate C (**49**) in 1995. Zhang et al. [14] also confirmed that the presence of matteuorienate A (**47**) in the rhizome of *M*. *struthiopteris*. Matteuorienates D–K (**50**–**57**) was found in the rhizome of *P. orientale* reported by Huh et al. [12]. From the *P*. *intermedium*, matteuorienate A (**47**), matteuorienate B (**48**), matteuorienate D (**50**), matteuorienate F (**52**), matteuorienate H (**54**), matteuorienate I (**55**), matteuorienate J (**56**), and matteuorienate K (**57**) were also isolated and reported by Li et al. [13]. In 2019, Li et al. [13] discovered matteuinterates A–F (**58**–**63**) along with matteuorienates, and these compounds have been isolated only from *P*. *Intermedium* so far. The chemical structures of flavonoid glycosides are depicted in Figure 4 and Figure 5, and Table 2.

#### 2.2.2. Stilbene Derivatives

Six stilbene derivatives (**64**–**69**) have been reported from the rhizomes of *P*. *orientale* and *M*. *struthiopteris*. The chemical structures of all stilbene derivatives are shown in Figure 6 and Table 3. Pinosylvin (**64**) and pinosylvic acid (**65**) were isolated from *P*. *orientale* and reported in 1995 [10]. Zhang et al. [14] also reported pinosylvin (**64**) from *M*. *struthiopteris*. In 2017, Song et al. [17] identified four stilbene derivatives, pinosylvic acid (**65**), resveratrolic acid (**66**), gaylussacin (pinosylvin 3-*O*-β-D-glucoside) (**67**), and resveratrolic acid 5-*O*-β-D-glucoside (**68**) originated from *P. orientale*. Matteucen J (**69**) was isolated from *P*. *orientale*, as reported by Zhu et al. [18], and the structure possesses dihydrostilbene moiety.

#### 2.2.3. Isocoumarins, Phthalides, and Chromone Derivatives

Shao et al. [19] confirmed that three isocoumarins (**70**–**72**) and two phthalides (**74** and **75**) isolated from the rhizome of *P*. *orientale*, and two enantiomers, (–)-matteucen A (70) and (+)-matteucen A (**71**), were successfully separated and reported, but (±)-matteucen B (**72**), (±)-matteucen C (**74**), and (±)-matteucen D (**75**) were reported as a mixture. Thunberginol C (**73**) was found in the rhizome of *M*. *struthiopteris* [16]. A chromone, leptorumol (**76**) [12], and chromone glycoside, matteucen I (**77**) [18], were isolated and determined from the rhizome of *P*. *orientale*. Matteucen I (**77**) possesses C-β-D-glucose at C-8 position and the α-L-rhamnose was connected through the hydroxyl group of C-2 position of the glucose. Matteuinterin B (**78**) was reported by Li et al. [20] in 2000, and is a compound featuring an HMG moiety attached to a chromone glycoside. The structures and classification of compounds **70**–**78** are depicted in Figure 6 and Table 3.

#### 2.2.4. Lignan Glycosides

In 2016, two diastereomeric neolignan glycosides, Matteustruthiosides A and B (**79** and **80**), were isolated and characterized from the aerial parts of *M*. *struthiopteris* [21]. Their structures and classification are depicted in Figure 7 and Table 3.

#### 2.2.5. Isoprenoids Derivatives

Eight isoprenoid derivatives were all reported from *P*. *intermedium*, including six sesquiterpenes, one sesquiterpene glycoside, and one iridoid glycoside. Matteuinterin A (**81**), kankanoside P (**88**), (3*S*,6*S*)-6,7-dihydroxy-6,7-dihydrolinalool 3-*O*-β-D-glucopyranoside (**82**), (6*R*,7*E*,9*R*)-9-hydroxy-4,7-megastigmadien-3-one 9-*O*-β-D-glucopyranoside (**83**), (6*S*,7*E*,9*R*)-9-hydroxy-4,7-megastigmadien-3-one 9-*O*-β-D-glucopyranoside (**84**), byzantionoside B (**85**), isodonmegastigmane I (**86**), and 9ξ-*O*-β-D-Glucopyranosyloxy-5-megastigmen-4-one (**87**) were found in the rhizomes of *P*. *intermedium* [20]. Matteuinterin A (**81**) possesses a gymnomitrane-type sesquiterpenoid skeleton with D-glucopyranoside, and this type of sesquiterpene is the first report from the pteridophytes. All chemical structures and classification are shown in Figure 7 and Table 3.

#### 2.2.6. Miscellaneous

Three miscellaneous compounds are depicted in Figure 7 and Table 3. Matteuinterin C (**89**) was revealed as phenolic glycosides and isolated from the rhizome of *P*. *intermedium* [20]. Matteuinterin C (**89**) possesses the p-hydroxy benzoic acid attached with two glycosides including D-glucopyranoside and L-rhamnopyranoside. From the fresh plant of *M*. *struthiopteris*, chlorogenic acid (**90**) and L-*O*-caffeoylhomoserine (**91**), a conjugated compound of caffeic acid and the amino acid L-homoserine, were isolated [22].

### 2.3. Pharmacological Activities

Isolated compounds from *P*. *orientale*, *P. intermedium*, and *M. struthiopteris* have been evaluated in several pharmacological studies so far, including antiviral activity for H1N1 A/PR/8/34 and H9N2 A/chicken/Korea/01210/2001, inhibitory effect for Prostaglandin E_2_ (PGE_2_) production, α-glucosidase, aldose reductase, and radical scavenging activity (Table 4).

#### 2.3.1. Antiviral Activity

Huh et al. [12], Zhu et al. [21], and Li et al. [15] conducted antiviral assays on the isolated compounds. Huh et al. [12] screened the isolated compounds for their neuraminidase inhibitory activities against H1N1 influenza virus. Among them, demethoxymatteucinol (**1**), matteucinol (**2**), matteucin (**3**), methoxymatteucin (**6**), and 3′-hydroxy-5′-methoxy-6,8-dimethylhuazhongilexone (**9**) showed inhibitory activities, and Huh et al. carried out further tests. The selected compounds were evaluated for neuraminidase inhibition and cytopathic effect inhibition against two influenza viruses, H1N1 A/PR/8/34 and H9N2 A/chicken/Korea/01210/2001. These five compounds (**1**, **2**, **3**, **6**, and **9**) exhibited neuraminidase inhibitory activities with IC_50_ values of 30.3 ± 3.0, 25.2 ± 2.4, 23.9 ± 3.0, 24.5 ± 1.5, and 24.4 ± 2.0 μM for H1N1 virus, respectively, and 31.3 ± 5.7, 27.2 ± 3.2, 24.1 ± 1.3, 24.6 ± 0.8, and 23.1 ± 1.7 μM for H9N2 virus, respectively. In addition, cytopathic effect inhibitory activities of five compounds with EC_50_ values were shown as 30.7 ± 2.0, 26.9 ± 1.3, 22.9 ± 2.0, 23.0 ± 3.4, and 21.4 ± 2.0 μM, respectively. The cytotoxicity of five compounds was also carried out using Madin-Darby canine kidney (MDCK) cells. Demethoxymatteucinol (**1**) showed moderate cytotoxicity, with a 50% cytotoxic concentration (CC_50_) value of 77.6 μM, whereas the other compounds (**2**, **3**, **6**, and **9**) exhibited no significant cytotoxicity.

Zhu et al. [21] evaluated the neuraminidase inhibitory effect of the neolignane glycosides, matteustruthiosides A (**79**) and B (**80**), against the H1N1 influenza virus, however both compounds showed inactivity.

Li et al. [15] also tested the neuraminidase inhibition assay for isolated compounds. Li et al. evaluated the cell viability with AlamarBlue assay to further assay, except for cytotoxic compounds. Ophiofolius A (**10**), matteflavoside G (**26**), and kaempferol-3-*O*-β-D-glucopyranoside (**34**) showed low cytotoxicity and inhibitory effect against H1N1 with EC_50_ values of 6.8 ± 1.1, 30.5 ± 1.0, and 72.8 ± 1.1 μM, respectively.

In summary, the Structure–activity relationship (SAR) analysis showed that glycosylation at specific positions such as C-7 can significantly enhance or reduce activity depending on the compound type. For kaempferol-type compounds, glycosides such as matteflavoside G (**26**) showed enhanced activity, while in *C*-methylated flavanones from *P*. *orientale*, aglycones were generally more potent.

#### 2.3.2. Anti-Inflammatory Activity

Li et al. [20] evaluated and reported the anti-inflammatory effects of isoprenoid glycosides, **82**–**87**, on PGE_2_ production in LPS-induced RAW 264.7 (murine microglial cells). Three concentrations (12.5, 25, and 50 μM) of compounds were pretreated for 1 h, followed by incuction with LPS (200 ng/mL) for 24 h. Li et al. [20] quantified the production of PGE_2_ using an enzyme immunoassay kit, and cell viability was assessed utilizing the (3-(4,5-dimethylthiazol-2-yl)-2,5-diphenyltetrazolium bromide) (MTT) assay. Among the tested isoprenoid glycosides, (3*S*,6*S*)-6,7-dihydroxy-6,7-dihydrolinalool 3-*O*-β-D-glucopyranoside (**82**) exhibited potent inhibitory effect against PGE_2_ production with an IC_50_ value of 17.8 ± 1.5 μM. 9ξ-*O*-β-D-Glucopyranosyloxy-5-megastigmen-4-one (**87**) also showed moderate inhibition, with an IC_50_ value of 30.3 ± 2.1 μM. Based on the SAR analysis, glycosides with a linear monoterpene backbone exhibited stronger PGE_2_ inhibitory activity compared to those with megastigmane or chromone skeletons.

#### 2.3.3. α-Glucosidase Inhibitory Activity

Li et al. [13] investigated the hypoglycemic potential of isolated compounds by examining their α-glucosidase inhibitory activity. Among the tested compounds, farrerol (**4**), matteucin (**3**), matteucinol (**2**), methoxymatteucin (**6**), cyrtominetin (**7**), and 3′-hydroxymatteucinol (**8**) showed more potent inhibitory activity than other flavonoid glycosides, with IC_50_ values ranging from 12.4 to 69.7 μM. Acarbose, used as a positive control, exhibited an IC_50_ value of 172.3 μM, indicating that these six flavonoids demonstrated more effective result compared to the positive control. Li et al. [13] noted that all active flavonoids shared a free 7-hydroxy group, and that the presence of *C*-methyl groups at the C-6 and C-8 positions, together with hydroxy or methoxy substitution on the B-ring, contributed to the enhanced α-glucosidase inhibitory effects. Among these active flavonoids, cyrtominetin (**7**) showed the most inhibitory activity. The SAR findings reveal that 3′, 4′-dihydroxy B-ring, *C*-methylated at C-6 and C-8, and hydroxyl group at C-7 are critical structural features for α-glucosidase inhibition among *C*-methylated flavanones.

#### 2.3.4. Aldose Reductase Inhibition

Basnet [10] reported five new C-methylated flavonoids from the rhizome of *P*. *orientale* and investigated for their aldose reductase activity. Among them, matteuorienate A (**58**), B (**59**), and C (**60**) showed as active as epalrestat, a positive control, in the presence of 1% BSA condition.

#### 2.3.5. Antioxidant Activity

Kimura et al. [22] tested the radical scavenging activity of chlorogenic acid (**90**) and L-*O*-caffeoylhomoserine (**91**) isolated from the rhizome of *M*. *struthiopteris* using two different assays, the chemiluminescence assay and the 2,2-diphenyl-1-picrylhydrazyl (DPPH) radical degradation assay. In the chemiluminescence assay, chlorogenic acid (**90**) and L-*O*-caffeoylhomoserine (**91**) exhibited an IC_50_ value of 0.31 ± 0.01 and 0.45 ± 0.05 mM, respectively. In addition, in the DPPH radical degradation assay, chlorogenic acid (**90**) and L-*O*-caffeoylhomoserine (**91**) showed an IC_50_ value of 0.13 ± 0.01 and 0.30 ± 0.00 mM, respectively. These results indicate that L-*O*-caffeoylhomoserine (**91**) has potent antioxidant activity.

#### 2.3.6. Hypoglycemic Activity

Basnet et al. [9] evaluated the hypoglycemic activity of three compounds, demethoxymatteucinol (**1**), matteucinol (**2**), and 2′-hydroxymatteucinol (**5**) isolated from the rhizome of *P*. *orientale* using a streptozotocin (STZ)-induced diabetic rat model. The compounds were administered intraperitoneally as a single dose of 100 mg/kg to diabetes-induced rats, and their effects were evaluated by measuring blood glucose levels at 6 and 24 h after administration. Among the tested compounds, 2′-hydroxymatteucinol (**5**) exhibited the most potent blood glucose-lowering effect at 6 h after administration. The authors further evaluated 2′-hydroxymatteucinol (**5**) through additional experiments using five intraperitoneal doses at 50 and 100 mg/kg, and measured blood glucose levels after the final dose of drug administration. At 50 mg/kg, 2′-hydroxymatteucinol (**5**) demonstrated a favorable hypoglycemic effect (28.7%), comparable to that of the positive control (30.7%), tolbutamide, administered at 100 mg/kg. Subsequently, 2′-hydroxymatteucinol (**5**) was orally administered twice a day at doses of 5, 10, 25, 50, and 100 mg/kg, and blood glucose levels were obtained 6 h after the final dose. Significant hypoglycemic effects were observed in the groups receiving 25 (22.0%), 50 (14.8%), and 100 (27.5%) mg/kg.

## 3. Conclusions and Future Perspectives

In this review, we comprehensively summarized the botany, phytochemistry, and pharmacologic activities of three Onocleaceae species: *P. orientale*, *P. intermedium*, and *M*. *struthiopteris*. A total of 91 compounds, including flavonoids, flavonoid glycosides, stilbene derivatives, isocoumarins, phthalides, chromone derivatives, lignan glycosides, and isoprenoid glycosides have been isolated and identified from these species. Notably, most flavonoids and flavonoid glycosides possess a *C*-methylated aromatic ring at the C-6 and/or C-8 positions, and some flavonoid glycosides are conjugated with (*S*)-HMG moieties. Pharmacological investigations have demonstrated that several isolated compounds exhibit antiviral, anti-inflammatory, hypoglycemic, aldose reductase inhibitory, and antioxidant activities. For instance, matteflavoside G (**26**) showed strong antiviral activity against the H1N1 influenza virus, suggesting the need for further testing against other influenza strains. Additionally, 2′-hydroxymatteucin (**5**) exhibited notable hypoglycemic effects, and further studies such as glucose uptake assays in 3T3-L1 adipocytes or HepG2 cells are recommended. Cyrtominetin (**7**) demonstrated good α-glucosidase inhibitory activity, and its evaluation in cellular glucose uptake models may further validate its potential. These findings highlight the potential of Onocleaceae plants as valuable sources of bioactive natural products.

Although research has been conducted on the chemical constituents and bioactivities of *P. orientale*, *P. intermedium*, and *M*. *struthiopteris*, there are many Onocleaceae family plants such as *Onoclea* and *Onocleopsis* that remain largely unexplored. In addition, the bioactivities of many isolated compounds have only been evaluated basic in vitro assays. Therefore, further studies are needed including mechanistic investigations, in vivo experiments, etc. Moreover, considering the characteristic features such as *C*-methylation on the A-ring, diverse hydroxy or methoxy substitute pattern on the B-ring, and HMG conjugation in these plants, further biosynthetic pathway studies could provide valuable insights into the diverse metabolites. In addition, a metabolomics study utilizing web-based platforms such as GNPS [23], MetaboAnalyst [24], and NPAnalyst [25], which are based on LC-MS data, could facilitate not only to the discovery of novel compounds structurally related to those summarized in this review but also to the exploration of bioactive molecules within the Onocleaceae family. We firmly believe that continued phytochemical and pharmacological research on the Onocleaceae family will not only contribute to the discovery of promising bioactive agents but also promote further research into the group of pteridophytes.

## 4. Methodology

The keywords used in the review are “Onocleaceae” “Matteuccia”, “Pentarhizidium”, “*Matteuccia struthiopteris*”, “*Matteuccia orientalis*”, “*Matteuccia intermedia*”, “*Pentarhizidium orientale*”, “*Pentarhizidium intermedium*”, “constituents”, “isolation” at the Web of Science, PubMed, Google Scholar, Scifinder. More than 150 publications were retrieved in this paper from January 1990 to March 2025 and of these, 19 papers were selected based on their relevance to phytochemistry and/or pharmacology. The ChemDraw 23.1.2 software was utilized to draw the chemical structures.

## Figures and Tables

**Figure 1 plants-14-01608-f001:**
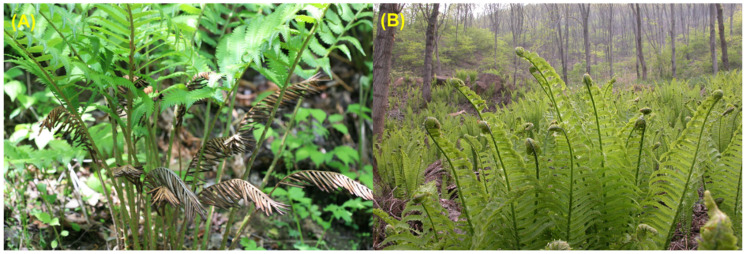
The aerial part of *Pentarhizidium orientale* (**A**), *Matteuccia struthiopteris* (**B**) (Figures (**A**,**B**) were adopted from “The National Insititute of Biological Resources” (https://species.nibr.go.kr) (accessed on 15 April 2025).

**Figure 2 plants-14-01608-f002:**
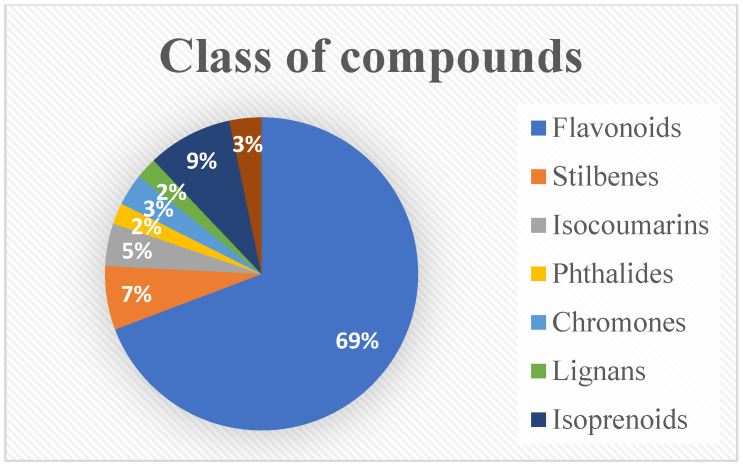
Distribution of compound classes identified from *P. orientale*, *P*. *intermedium*, and *M*. *struthiopteris*.

**Figure 3 plants-14-01608-f003:**
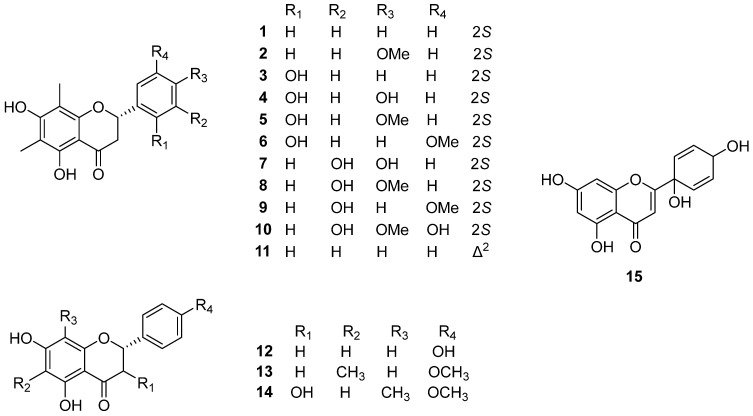
Chemical structures of flavonoids.

**Figure 4 plants-14-01608-f004:**
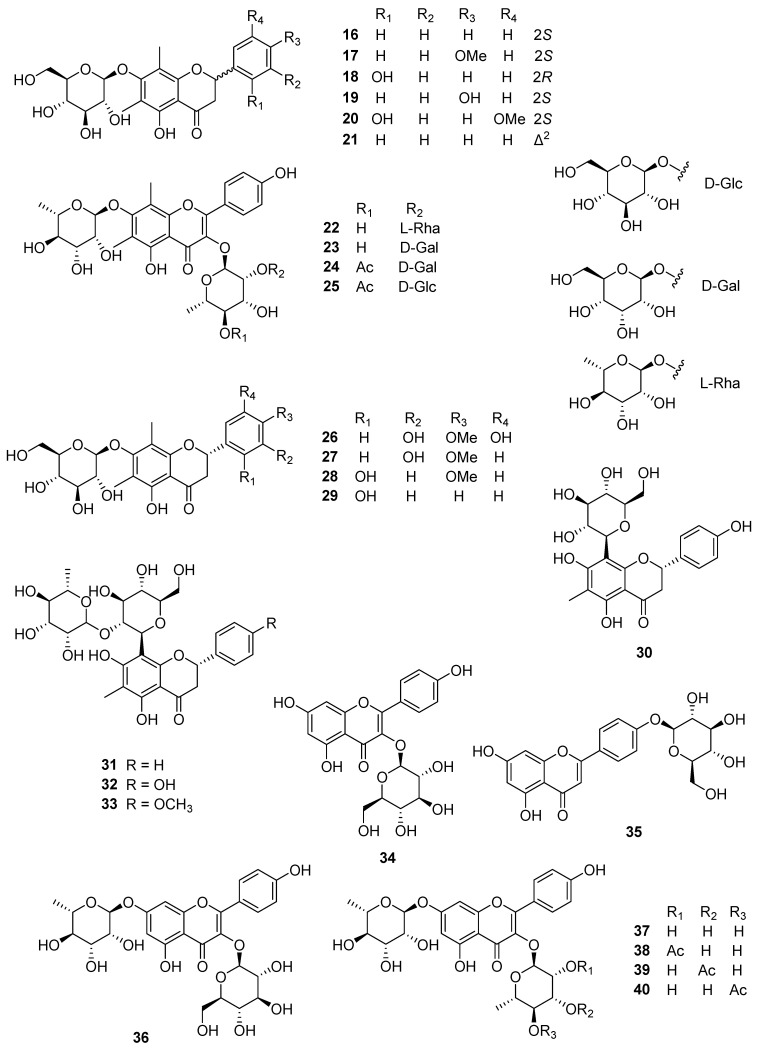
The chemical structures of flavonoid glycosides (**16**–**40**).

**Figure 5 plants-14-01608-f005:**
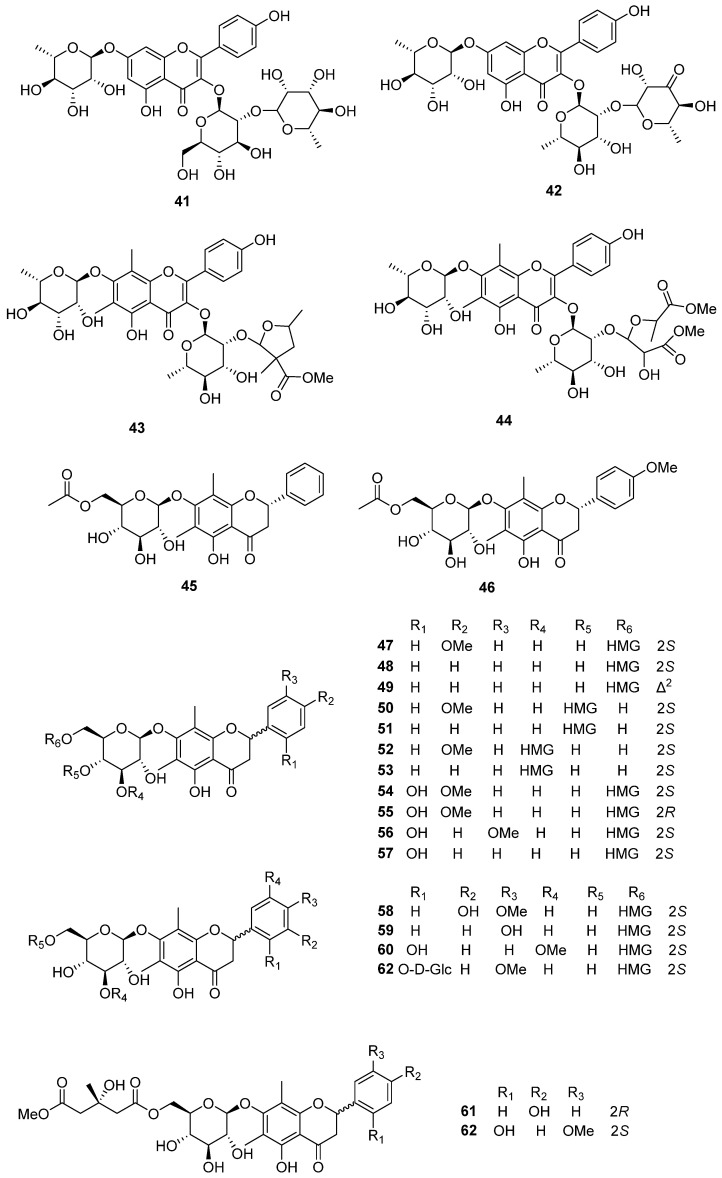
The chemical structures of flavonoid glycosides (**41**–**62**).

**Figure 6 plants-14-01608-f006:**
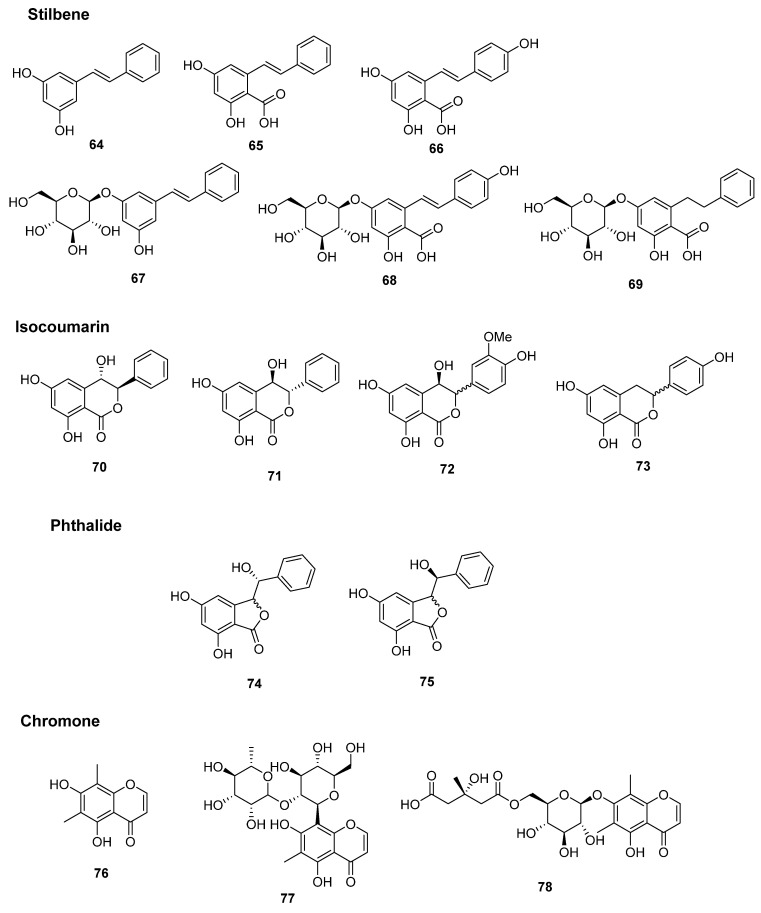
Chemical structures of stilbenes, isocoumarins, phthalides, and chromone derivatives.

**Figure 7 plants-14-01608-f007:**
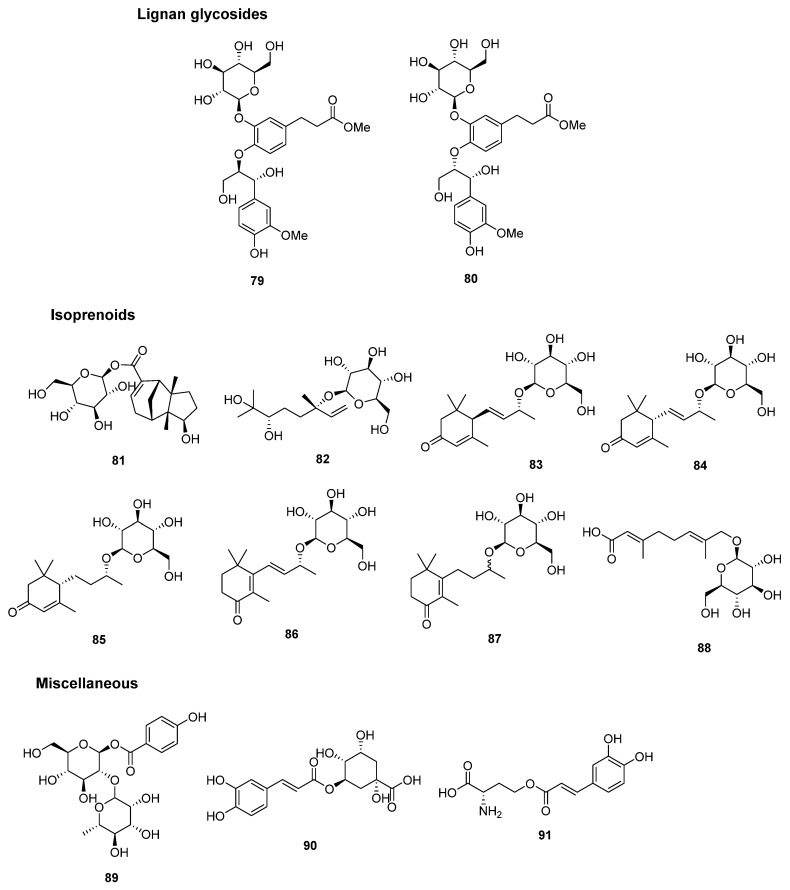
The chemical structure of lignan glycosides, isoprenoid glycosides, and miscellaneous compounds.

**Table 1 plants-14-01608-t001:** Flavonoids from *P*. *orientale*, *P*. *intermedium*, and *M*. *struthiopteris*.

No.	Compound Name	Part of the Plant	Plant Species	References
Flavonoids				
1	Demethoxymatteucinol	rhizome, root	PO	[8,9,10,11,12]
		rhizome	PI	[13]
		rhizome	MS	[14]
2	Mattuecinol	rhizome, root	PO	[8,9,10,11,12]
		rhizome	PI	[13]
		rhizome	MS	[14]
3	Matteucin	rhizome, root	PO	[8,12,13]
			PI	[13]
4	Farrerol	rhizome	PO	[12]
			PI	[13]
5	2′-Hydroxymatteucinol	rhizome	PO	[9,10,11]
6	Methoxymatteucin	rhizome, root	PO	[8,12]
			PI	[13]
			MS	[16]
7	Cyrtominetin	rhizome	PI	[13]
8	3′-Hydroxymatteucinol	rhizome	PI	[13]
9	3′-Hydroxy-5′-methoxy 6,8-dimethyl huazhongilexone	rhizome	PO	[12]
10	Ophiofolius A	rhizome	MS	[15]
11	Matteuorien	rhizome	PO	[10]
			MS	[15]
12	Naringenin	rhizome	PO	[12]
PO	5,7-Dihydroxy-4′-methoxy-6-methyl flavanone	rhizome	PI	[13]
			MS	[16]
14	Demethylmatteucinol	rhizome	PI	[13]
15	Protoapigenone	rhizome	MS	[15]

PO: *P. orientale*; PI: *P. intermedium*; MS: *M. struthiopteris*.

**Table 2 plants-14-01608-t002:** Flavonoid glycosides from *P*. *orientale*, *P*. *intermedium*, and *M*. *struthiopteris*.

No	Compound Name	Part of the Plant	Plant Species	References
Flavonoid glycosides				
16	Demethoxymatteucinol 7-*O*-β-D-glucoside	rhizome	PO	[10,12]
			PI	[13]
			MS	[16]
17	Mattuecinol 7-*O*-β-D-glucopyranoside	rhizome	PO	[10,12]
			PI	[13]
			MS	[16]
18	Matteucin 7-*O*-β-D-glucopyranoside	rhizome	PO	[12]
19	Farrerol 7-*O*-β-D-glucopyranoside	rhizome	PI	[13]
20	Myrciacitrin II	rhizome	PO	[12]
			PI	[13]
21	Matteuorien 7-*O*-β-D-glucopyranoside	rhizome	PO	[12]
22	Matteflavoside A	rhizome	MS	[15]
23	Matteflavoside B	rhizome	MS	[15]
24	Matteflavoside C	rhizome	MS	[15]
25	Matteflavoside D	rhizome	MS	[15]
26	Matteflavoside G	rhizome	PI	[13]
			MS	[15]
27	Matteflavoside H	rhizome	PI	[13]
28	Matteflavoside I	rhizome	PI	[13]
29	Matteflavoside J	rhizome	PI	[13]
30	Matteuorienin	rhizome	PO	[12]
31	Matteuorienin B	rhizome	PO	[12]
32	Matteuorienin C	rhizome	PO	[12]
33	Matteuorienin D	rhizome	PO	[12]
34	Kaempferol-3-*O*-β-D-glucopyranoside	rhizome	MS	[15]
35	Apigenin-4′-*O*-β-D-glucopyranoside	rhizome	MS	[15]
36	Kaempferol-3-*O*-β-D-glucopyranosyl-7-*O*-α-L-rhamnopyranoside	rhizome	MS	[15]
37	Kaempferol-3,7-di-*O*-α-L-rhamnopyranoside	rhizome	MS	[15]
38	Kaempferol-3-*O*-(α-L-2-*O*-acetyl-Khamnopyranosyl)-7-*O*-α-L-rhamnopyranoside	rhizome	MS	[15]
39	Kaempferol-3-*O*-(α-L-3-*O*-acetyl-rhamnopyranosyl)-7-*O*-α-L-rhamnopyranoside	rhizome	MS	[15]
40	Kaempferol-3-*O*-(α-L-4-*O*-acetyl-rhamnopyranosyl)-7-*O*-α-L-rhamnopyranoside	rhizome	MS	[15]
41	Kaempferol-3-*O*-[β-D-glucopyranosyl-(1→2)-α-L-rhamnopyranosyl]-7-*O*-α-L-rhamnopyranoside	rhizome	MS	[15]
42	Kaempferol-3-*O*-[1,2,4-trihdroxy-3-oxo-5-methyltetrahydropyran-(1→2)-α-L-rhamnopyranosyl]-7-*O*-α-L-rhamnopyranoside	rhizome	MS	[15]
43	Matteflavoside E	rhizome	MS	[15]
44	Matteflavoside F	rhizome	MS	[15]
45	(2*S*)-5,7-dihydroxy-6,8-dimethyl-dihydroflavone-7-*O*-(6″-*O*-acetyl)-β-D-glucopyranoside	rhizome	MS	[16]
46	(2*S*)-5,7-dihydroxy-6,8-dimethyl-4′-methoxydihydroflavone-7-*O*-(6″-*O*-acetyl)-β-D-glucopyranoside	rhizome	MS	[16]
47	Mateuorienate A	rhizome	PO	[10,11,12]
			PI	[13]
			MS	[14]
48	Mateuorienate B	rhizome	PO	[10,11,12]
			PI	[13]
49	Mateuorienate C	rhizome	PO	[10,12]
50	Mateuorienate D	rhizome	PO	[12]
			PI	[13]
51	Mateuorienate E	rhizome	PO	[12]
52	Mateuorienate F	rhizome	PO	[12]
			PI	[13]
53	Mateuorienate G	rhizome	PO	[12]
54	Mateuorienate H	rhizome	PO	[12]
			PI	[13]
55	Mateuorienate I	rhizome	PO	[12]
			PI	[13]
56	Matteuorienate J	rhizome	PO	[12]
			PI	[13]
57	Matteuorienate K	rhizome	PO	[12]
			PI	[13]
58	Matteuinterate A	rhizome	PI	[13]
59	Matteuinterate B	rhizome	PI	[13]
60	Matteuinterate C	rhizome	PI	[13]
61	Matteuinterate D	rhizome	PI	[13]
62	Matteuinterate E	rhizome	PI	[13]
63	Matteuinterate F	rhizome	PI	[13]

PO: *P. orientale*; PI: *P. intermedium*; MS: *M. struthiopteris*.

**Table 3 plants-14-01608-t003:** Several classes of compounds from *P*. *orientale*, *P*. *intermedium*, and *M*. *struthiopteris*.

No	Compound Name	Part of Plant	Plant	References
Stilbene derivatives				
64	Pinosylvin	rhizome	PO	[10]
			MS	[14]
65	Pinosylvic acid	rhizome	PO	[10,17]
66	Resveratrolic acid	rhizome	PO	[17]
67	Gaylussacin (Pinosylvin 3-*O*-β-D-glucopyranoside)	rhizome	PO	[17]
			MS	[14]
68	Resveratrolic acid 5-*O*-β-D-glucopyranoside	rhizome	PO	[17]
69	Matteucen J	rhizome	PO	[18]
Isocoumarins				
70	(-)-matteucen A	rhizome	PO	[19]
71	(+)-matteucen A	rhizome	PO	[19]
72	(±)-matteucen B	rhizome	PO	[19]
73	Thunberginol C	rhizome	MS	[16]
Phthalides				
74	(±)-Matteucen C	rhizome	PO	[19]
75	(±)-Matteucen D	rhizome	PO	[19]
Chromone derivatives				
76	Leptorumol	rhizome	PO	[12]
77	Matteucen I	rhizome	PO	[18]
78	Matteuinterin B	rhizome	PI	[20]
Lignan glycosides				
79	Matteustruthioside A	aerial parts	MS	[21]
80	Matteustruthioside B	aerial parts	MS	[21]
Isoprenoid derivatives				
81	Matteuinterin A	rhizome	PI	[20]
82	(3*S*,6*S*)-6,7-dihydroxy-6,7-dihydrolinalool 3-*O*-β-D-glucopyranoside	rhizome	PI	[20]
83	(6*R*,7*E*,9*R*)-9-hydroxy-4,7-megastigmadien-3-one 9-*O*-β-D-glucopyranoside	rhizome	PI	[20]
84	(6*S*,7*E*,9*R*)-9-hydroxy-4,7-megastigmadien-3-one 9-*O*-β-D-glucopyranoside	rhizome	PI	[20]
85	Byzantionoside B	rhizome	PI	[20]
86	isodonmegastigmane I	rhizome	PI	[20]
87	9ξ-*O*-β-D-Glucopyranosyloxy-5-megastigmen-4-one	rhizome	PI	[20]
88	Kankanoside P	rhizome	PI	[20]
Miscellaneous				
89	Matteuinterin C	rhizome	PI	[20]
90	Chlorogenic acid	rhizome	MS	[22]
91	L-*O*-caffeoyl-homoserine	rhizome	MS	[22]

PO: *P. orientale*; PI: *P. intermedium*; MS: *M. struthiopteris*.

**Table 4 plants-14-01608-t004:** Pharmacological activities of isolated compounds.

No	Compounds	Study Model	Dose and/or Concentration	Effects	References
Antiviral activity					
1	Demethoxymatteucinol	H1N1, A/PR/8/34 H9N2, A/chicken/Korea/01210/2001	IC_50_: 30.3 μM/ 31.3 μM EC_50_: 30.7 μM	neuraminidase inhibitory activity	[12]
2	Matteucinol		IC_50_: 25.2 μM/ 27.2 μM EC_50_: 26.9 μM		
3	Matteucin		IC_50_: 23.9 μM/ 24.1 μM EC_50_: 22.9 μM		
6	Methoxymatteucin		IC_50_: 24.5 μM/ 24.6 μM EC_50_: 23.0 μM		
9	3′-hydroxy-5′-methoxy 6,8-dimethyl huazhongilexone		IC_50_: 24.4 μM/ 23.1 μM EC_50_: 21.4 μM		
10	Ophiofolius A	H1N1	EC_50_: 72.8 μM	neuraminidase inhibitory activity	[15]
26	Matteflavoside G		EC_50_: 6.8 μM		
34	Kaempferol-3-*O*-β-D-glucopyranoside		EC_50_: 30.5 μM		
Anti-inflammatory activity					
82	(3*S*,6*S*)-6,7-dihydroxy-6,7-dihydrolinalool 3-*O*-β-D-glucopyranoside	LPS-induced RAW 264.7 murine macrophages	IC_50_: 17.8 μM	PGE_2_ production inhibition	[20]
83	(6*R*,7*E*,9*R*)-9-hydroxy-4,7-megastigmadien-3-one 9-*O*-β-D-glucopyranoside		IC_50_: > 50 μM		
84	(6*S*,7*E*,9*R*)-9-hydroxy-4,7-megastigmadien-3-one 9-*O*-β-D-glucopyranoside		IC_50_: > 50 μM		
85	Byzantionoside B		IC_50_: > 50 μM		
86	Isodonmegastigmane I		IC_50_: > 50 μM		
87	9ξ-*O*-β-D-Glucopyranosyloxy-5-megastigmen-4-one		IC_50_: 30.3 μM		
α-Glucosidase activity					
2	Matteucinol	In vitro, Enzyme (0.5 U/mL) from *Saccharomyces cerevisiae*), substrate: *p*-NPG	IC_50_: 28.0 μM	α-Glucosidase inhibition	[13]
3	Matteucin		IC_50_: 37.6 μM		
4	Farrerol		IC_50_: 44.1 μM		
6	Methoxymatteucin		IC_50_: 69.7 μM		
7	Cyrtominetin		IC_50_: 12.4 μM		
8	3′-hydroxymatteucinol		IC_50_: 43.6 μM		
Aldose reductase inhibition					
47	Matteuorienate A	In vitro, Enzyme (eye lens from 5-week-old male Wistar rats) Substrate: glyceraldehyde	IC_50_: 3.6 μM (in presence of 1% BSA)	Aldose reductase inhibition	[10]
48	Matteuorienate B		IC_50_: 3.7 μM (in presence of 1% BSA)		
49	Matteuorienate C		IC_50_: 6.4 μM (in presence of 1% BSA)		
Antioxidant activity					
90	Chlorogenic acid	Chemiluminescence method, DPPH radical degradation method	IC_50_: 0.31 μM IC_50_: 0.13 μM	Radical scavenging activity	[22]
91	L-*O*-caffeoylhomoserine		IC_50_: 0.45 μM IC_50_: 0.30 μM		
Hypoglycemic activity					
5	2′-hydroxymatteucinol	In vivo, STZ-induced diabetic rats	i.p.: 28.7% (50 mg/kg)/ 38.2% (100 mg/kg) p.o.: 22.0% (25 mg/kg)/ 14.8% (50 mg/kg)/ 27.5% (100 mg/kg)	Reducing blood glucose level	[9]

LPS: Lipopolysaccharide; IC_50_: half-maximal inhibitory concentration. BSA: Bovine serum albumin; IC_50_: half-maximal inhibitory concentration; *p*-NPG: *p*-nitrophenyl glucopyranoside; DPPH: 2,2-diphenyl-1-picrylhydrazyl; STZ: streptozotocin; p.o.: per os; i.p.: intraperitoneal.

## Data Availability

Not applicable.

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
