# Peer review of "The Phytochemistry and Pharmacology of Onocleaceae Plants: Pentarhizidium orientale, Pentarhizidium intermedium, and Matteuccia struthiopteris—A Review"

_plants, 2025, doi:10.3390/plants14111608_

Round 1

Reviewer 1 Report

Comments and Suggestions for Authors

The review entitled “The phytochemistry, pharmacology of Onocleaceae plants; Pentarhizidium orientale, Pentarhizidium intermedium, and Matteuccia struthiopteris: A review”  presents a brief summarising of the botany, phytochemistry and pharmacological activities of three Onocleaceae species: P. orientale, P. intermedium, and M. struthiopteris. The review is accurately  written, actual, presents the new analysis of botany, phytochemistry and pharmacological activities of Onocleaceae species. For example, a group of 91 compounds have been isolated from these  species, including 15 flavonoids, 48 flavonoid glycosides, 6 stilbenes, 4 isocoumarins, 2 phthalides, 3 chromones, 2 lignan glycosides, 8 isoprenoid derivatives, and 3 phenolic compounds. Several isolated compounds exhibit antiviral, anti-inflammatory, hypoglycemic and anti-oxidant activities. The topic of the review is original and possess novelty since the search of new natural extracts as valuable source of boactive compounds is needed. The main question addressed by this research is the analysis of phytochemistry and pharmacological activities of three Onocleaceae species. The conclusions consistent with the evidence and are detailed, the necessary arguments are presented and addressed to the main question posed. The figures and tables are concisely and clearly reflect the details of the discussed topic. The paper could be accepted to Plants after revision. The recommendations to the author are the following: 1) is their any information about the influence of place of growth or other conditions on the compound’s content? 2) please indicate any differences in the phytochemistry or pharmacological activities of these species with similar ones; 3) please indicate any relation in the structure of the isolated compound and the type of  activity; 4) Line 215 – please add “activity” 5) please revise Table 4 : for ex, for compounds 1-3, 6, 9 the Study Model and Effects are similar . The same truth for 26, 34 , 10 and other ones.

Author Response

I am grateful to the reviewer for their careful reading of my manuscript and for providing constructive feedback. I have thoroughly addressed all comments and revised the manuscript accordingly.

1) is their any information about the influence of place of growth or other conditions on the compound’s content?

→ Thank you for your good question. As you pointed out, I also aimed to gather information on how the place of growth and other conditions affect the compound content; however, it was difficult to find relevant data. I believe that as more research is conducted on plants of Onocleaceae family in the future, it will become possible to systematically summarize the influence of factors such as collection site, climate, and etc.

2) please indicate any differences in the phytochemistry or pharmacological activities of these species with similar ones;

→ Thank you for your comment. As you pointed out, it would be valuable to compare the phytochemistry or pharmacological activities of these species with related ones. However, as noted in the Introduction section, other plants within the Onocleaceae family have not been sufficiently studied, making such comparisons difficult. Furthermore, expanding the scope to the broader Polypodiales order, which includes a large number of families, would go beyond the intended focus of this study. In the future, if more studies are conducted on other plants within the Onocleaceae family, it would be possible to compare differences in phytochemistry or pharmacological activity, as you suggested. Thank you again for your valuable comment.

3) please indicate any relation in the structure of the isolated compound and the type of activity;

→ Thank you for your comment. As you suggest, I conducted SAR analyses for the antiviral, anti-inflammatory, and α-glucosidase inhibitory activities.

4) Line 215 – please add “activity”

→ Thank you for the comment. I added the “activity” in the manuscript.

5) please revise Table 4: for ex, for compounds 1-3, 6, 9 the Study Model and Effects are similar. The same truth for 26, 34, 10 and other ones.

→ Thank you for your comment. I revised Table 4.

Reviewer 2 Report

Comments and Suggestions for Authors

Please refer to the attached file for comments/suggestions. 

Comments on the Quality of English Language

There are a few typo errors in the text, refer to the comments and correct them. 

Author Response

I am grateful to the reviewer for their careful reading of my manuscript and for providing constructive feedback. I have thoroughly addressed all comments and revised the manuscript accordingly.

  1. Italicize the names of plants in the title and keywords.

→ Thank you for your comment. I corrected as you mentioned.

  1. In line 66, check spelling “frons or fronds”?

→ Thank you, it was a typo. It has been corrected to Fronds

  1. Line 94, compound (5) not bolded.

→ Corrected

  1. Lines 147-157, compound numbers in the bracket were not bolded, unlike rest of the text.

→ Thank you. All the numbers in the bracket have been bolded.

  1. Line 147-157, plant names not italicized.

→ Corrected.

  1. Line 188, give full forms of H1N1, H9N1, and PGE2, as they appear first time in the text. Although author has given the list of abbreviations, it would be better to give full forms wherever it first appears in the text.

→ Corrected. The full names of H1N1, H9N1, and PGE2 have been added at their first mention in the manuscript

  1. Line 204, what is MDCK cells, and in line 205, what is CC50? If author could give full form of new abbreviations for non-technical audience/readers.

→ As with comment 6, the full names of MDCK and CC50 have been added in the manuscript at their first mention, and the abbreviation have also been included in the abbreviation section.

  1. Author used full stop after “et al.” but not in some cases, make it uniform throughout the text.

→ Thank you. I have thoroughly reviewed the entire manuscript and standardized the usage to “et al.” throughout.

  1. In line 220, what does MTT assay stands for non-technical readers?

→ Corrected. The full form of MTT has been provided in the manuscript.

  1. Line 238, plant name not italicized.

→ Corrected.

  1. Lines 238-241, compound numbers not in bold.

→ Corrected.

  1. Line 245, what is full form of DPPH?

→ Thank you. As with the response to comment 9, the corrected revisions have been made.

  1. Lines 256-257, revise the sentence. Does it mean “Notably, almost all flavonoids and ……”?

→ Corrected.

  1. Under method section, italicize all scientific names of plants. Similarly, cross check throughout the text, and make sure all scientific names of plants are italicized, e.g., in Table 1 title and footnote as well.

→ The scientific names of the plants in the method section have all been italicized. In addition, all scientific names appearing in the main text and tables have also been corrected to italics.

  1. In table 4, if author could add full forms for all cells/study models and for any abbreviated words used in the table as table footnote.

→ The revisions have been made according to the reviewer’s comment.

  1. In the conclusion, if author could highlight/mention few promising bioactive compounds identified from the review and recommend further research on those promising compounds. Were there any compounds (or novel compounds) isolated from the plants but not tested for biological activity? If so, why they were not tested? Do you have any suggestions?

→ Thank you for the thoughtful question. Among the compounds with antiviral activity, the flavonoid glycoside matteflavoside G (26) showed strong activity against the H1N1 influenza virus; therefore, I recommend testing it against other influenza virus strains as well. In the evaluation of hypoglycemic activity, 2’-hydroxymatteucin (5) exhibited notable effects, and I would suggest further experiments such as the glucose uptake assays using 3T3-L1 adipocytes or in HepG2 cells. Additionally, cyrtominetin (7) demonstrated good inhibitory activity in the α-glucosidase assay. As mentioned above, it may be worthwhile to conduct glucose uptake assays in 3T3-L1 adipocytes or HepG2 cells for further evaluation. These points have been incorporated into the revised manuscript.

 Most of the isolates were evaluated for their biological activity. However, some could not be tested due to the limited quantity available, which is a common challenge faced by natural product researchers. In addition, a few compounds were reported without biological evaluation due to the publication scope of the journals in which they appeared. Although it is difficult to provide a definitive solution to this issue, I hope that in the future, sufficient quantities of these compounds can be obtained for activity testing, and the growing scientific interest in these plants will lead to further investigations not only of their chemical structures but also of their biological activities.

  1. In the conclusion, author mentioned that “metabolomics study could facilitate new research opportunities within Onocleaceae family”? Could you be specific, what kind of opportunity, whether it is going to ease the drug discovery process or something else?

→ Thank you for your comment. What I meant is that when conducting future studies on other plants within the Onocleaceae family, metabolomics research using LC-MS data and web-based platforms could potentially lead to the discovery of new compounds that are structurally similar to those summarized in this review, but also to the exploration of bioactive compounds. To convey this point more clearly, I have revised the sentence in the manuscript.

  1. In the Table 4 (biological activity of compounds), list is very short for the biological activity (and mostly in vitro studies, not much in vivo studies). Was this a random and selective search or did you do thorough search and found only this much?

→ I conducted the review after searching as many publications as possible using the keywords provided in the Methods section. Most of the studies conducted in vitro experiments, and in vivo studies were difficult to find. Only one study reported in vivo experiments, and its details have been added to the manuscript and table. Some papers reported in vitro experiments of extracts or conducted peak profiling using LC-MS or GC-MS; however, these were excluded from the review because the studies did not isolate compounds or evaluated the bioactivity of the isolates.